# Observations on Refractive Status and Risk Factors for Visual Impairment in Children with Disabilities

**DOI:** 10.3390/medicina57050403

**Published:** 2021-04-22

**Authors:** Larisa Bianca Holhoș, Mihaela Cristiana Coroi, Liviu Lazăr

**Affiliations:** 1Department of Morphological Disciplines, Faculty of Medicine and Pharmacy, University of Oradea, 10 December Square Street, 410028 Oradea, Romania; 2Medical Doctoral School, University of Oradea, 1 Universitatii Street, 410087 Oradea, Romania; 3Department of Ophthalmology, Emergency County Hospital, 37 Republicii Street, 410087 Oradea, Romania; 4Department of Surgical Disciplines, Faculty of Medicine and Pharmacy, University of Oradea, 10 December Square Street, 410028 Oradea, Romania; 5Department of Psycho-Neuroscience and Recovery, Faculty of Medicine and Pharmacy, University of Oradea, 10 December Square Street, 410028 Oradea, Romania; lazarlv@yahoo.com

**Keywords:** children, special health needs, refractive errors, visual impairment, risk factors, vitamin D level

## Abstract

Vision integrates all the senses and plays a fundamental role in the acquisition of different skills and the general development of a child. Recently, refractive errors and visual impairment have become serious health problems among children. *Background and Objectives*: The aim of this study was to identify the prevalence of refractive errors and visual impairment in children with disabilities. Moreover, this study aimed to explore the risk factors for visual impairment in children with respect to vitamin D levels, parental smoking, and the use of spectacles. *Materials and Methods*: We retrospectively analyzed 161 children aged 5–16 years old, divided into two groups: a group of 80 children with disabilities and a control group consisting of 81 clinically healthy children. All the children underwent refraction measurements, visual acuity testing, and plasmatic vitamin D level dosing, measured in ng/mL. *Results*: Refractive errors and visual impairment were found to have a higher prevalence in the group of children with disabilities than in the control group. Moreover, the plasma level of vitamin D was lower in the group of children with special health needs. *Conclusions*: Given the present results, children with disabilities should undergo special eye examinations as soon as possible to ensure the quality of their socio-academic lives.

## 1. Introduction

There are about 1.4 million children globally who face the challenges of being diagnosed with refractive errors [1]. Up to 43% of these children have a visual dysfunction, making refractive errors a public health problem [2]. Moreover, uncorrected refractive errors are considered to be one of the most debilitating disabilities a child can have [3]. Due to this disability, the cognitive development of the child and his or her socio-academic status can be negatively impacted compared to children who do not have such a condition [4,5]. Visual impairment can manifest as a reduction in visual acuity and loss of the visual field, leading to impacts on general development. Visual impairment can be caused by untreated refractive errors, cataracts, and glaucoma. There are several studies that outline the risk factors for refractive errors and visual impairment, including genetic and environmental factors such as prematurity, genetic syndromes, different disabilities, a former or present smoking mother or father, and parents who wear glasses [6].

On 9 June 2011, the World Report on Disability (WRD) was released. The WRD states that a disability refers to any obstacle that interferes with the environment of a person with a condition [7]. On 31 March 2020, the National Authority for Disabled People reported 852.565 persons registered with disabilities in Romania, among whom 68.447 were children [8]. Law no. 448/2006 outlines the existence of different types of disabilities or special health needs, such as mental, somatic, visual, and rare diseases. Some of the conditions listed as disabilities include Down syndrome, dyslexia, autism, blindness, and different motor conditions. These children face many obstacles, so they need assistance in handling their conditions [8].

Studies conducted on the pediatric population showed this population to have various ocular diseases and refractive errors at a higher prevalence than children without disabilities. Refractive errors are identified to occur with a prevalence of between 10.5 and 57% among children with special health needs, but many conducted studies have only been descriptive and lacked a control group [9]. These children often cannot express themselves properly and, despite engaging in physical-, occupational-, and communication-based therapy, often do not receive an ophthalmological check-up. Children shape their lives around their visual information, and, due to their different deficiencies, eye problems and secondary visual impairments must be detected and corrected as soon as possible. Thus, visual screening is recommended for children before they enter school in order for them to succeed academically and achieve a good quality of life.

The International Council of Ophthalmology (ICO) recommends classifying vision loss based on visual acuity, where normal vision refers to visual acuity ≥0.8 and visual impairment or vision loss begins when one has a visual acuity <0.8. This classification is suitable for clinical research and population studies [10]. Some studies have observed the development of emmetropization based on adequate plasma levels of vitamin D [11]. This vitamin is considered a multifunctional hormone, which eye tissues can activate and respond to. Epidemiological studies have reported that low plasmatic levels can influence refractive status and visual status [12]. Special needs children must undergo an ophthalmological examination because they are prone to refractive disorders. Once such a disorder is discovered, the child can be treated and his or her quality of life can be improved. The refractive status and visual acuity of children can be influenced by environmental factors such as sun exposure, secondary plasmatic vitamin D levels, and exposure to tobacco smoke. Risk factors for visual impairment among children include parents’ refractive status and smoking habits [13,14].

This study aimed to assess the refractive status of children with disabilities and to compare the results with those of a control group consisting of clinically healthy children. The plasmatic vitamin D status of all the children included in the study was dosed in order to correlate that status with the initial visual acuity of all the children. Another aim of the study was to analyze the presence of risk factors for the development of visual impairment and to calculate a relative risk for its development, based on the parents’ smoking status and wearing of glasses.

## 2. Materials and Methods

### 2.1. Study Subjects

In this research, a retrospective case-control study was conducted according to the Declaration of Helsinki. Ethical clearance no. 14/23.12.2020 was obtained from the Ethics and Research Committee of the Faculty of Medicine and Pharmacy of Oradea, University of Oradea, Bihor county, Romania.

In the study, we included 161 children for statistical analysis. The children were divided into two groups: 80 children who had been previously diagnosed with disabilities and 81 clinically healthy children. The children were examined from January 2019 to August 2020 at a private general practitioner’s office (Elisantes Teola SRL) from Marghita, Bihor county, Romania. The inclusion criteria for the children were as follows: (1) children diagnosed with disabilities; (2) clinically healthy children with no history of any disability or chronic diseases; and (3) children aged between 5 and 16. The reference group consisted of children with disabilities whose diagnoses were established based on medical documents given by the parents to the eye care specialists on the day of the consultation. The types of disabilities identified among the children in this study group included Down syndrome, autism, ADHD, deafness, scoliosis, epilepsy, an intellectual disability, motor deficits, phocomelia, dyslexia, and agenesis of the superior hand’s fingers. The group of children without disabilities was established based on documents from each child’s current general practitioner stating that the child was clinically healthy and free from any psychological or physical pathology.

All the children underwent ocular examinations, including refraction measurements and the testing of visual acuity. We measured the initial visual acuity without any correction and the final visual acuity with or without correction, depending on the case. All parents of the children included in this study were informed of the stages of the study and provided signed consent to take part in the study for both the ophthalmological examination and vitamin D dosing. Each stage of the examination was explained to the children in an appropriate manner, allowing the children to be more relaxed and attentive.

### 2.2. Refraction Measurement

The ophthalmological assessment was performed in a 5 m length room, after the procedure was fully explained to the children and their parents. Visual acuity was measured initially for each eye, without correction, using a letter or Lea symbol decimal scale followed by the appropriate correction according to the cycloplegic refraction measurement on the same day, if possible, or on another day. Cycloplegia was tested with cyclopentolate hydrochloride drops (SC Rompharm Company SRL, Otopeni, Romania) three times at ten-minute intervals. Drops were instilled into the inferior conjunctival sac after the parents’ confirmation that their children had not experienced any seizures in the past. Each drop of cyclopentolate hydrochloride contained 0.3 mg of active substance. Cycloplegic refraction was tested thirty minutes after the moment that the third drop of cyclopentolate was instilled into the inferior conjunctival sacs of both eyes. Cycloplegia was diagnosed when a minimum 6 mm pupil diameter was achieved or in the absence of a pupillary light reflex. Refraction measurements were carried out using a Canon Full Auto Ref-Keratometer RK-F2 (Canon INC Kawasaki, Kanagawa, Japan, 2016) for each eye.

### 2.3. Vitamin D Dosing

After signing the consent for vitamin D dosing, the parents of the children were asked to dose their children with vitamin D at any desired local laboratory. Determination of the serum level of 25-OH vitamin D was completed for all children via the immunochemical method with electro-chemiluminescent detection; this method measures the total level of vitamin D, which consists of vitamin D3, vitamin D2, and other hydroxylated metabolites of vitamin D. All the local laboratories had the same reference range values.

### 2.4. Systemic Factors

We collected data on the demographic factors of each child enrolled in the study, such as age, gender type, residence of the child, risk factors such as having a mother and/or father who formerly smoked, a mother and/or father who currently smokes, a mother and/or father who never smoked, and the medical history of the parents (e.g., wears glasses).

### 2.5. Statistical Analysis

Statistical analysis was performed using the medical statistics program MedCalc version 12.5.0.0 (MedCalc Software, Mariakerke, Belgium). Categorical data are expressed as absolute numbers and continuous data as the mean ± standard deviation (SD) (95% confidence interval) or as the median (interquartile range, IQR). We used a Mann–Whitney test for non-normal distribution. The variables were analyzed using a chi-square test with Yates’ correction for continuity and Student’s *t*-test for independent groups. A chi-square test was used to compare the categorical data. Correlations were evaluated using the Spearman coefficients, as appropriate. The vitamin D level presented normal distribution, while the initial visual acuity featured asymmetric distribution (skewed). A Spearman test was used to assess the correlation between the vitamin D level and the initial visual acuity. The relative visual impairment risk was determined using a logistic regression test with gradual entry by calculating the odds ratio (OR) with a confidence interval of 95%. All *p* values < 0.05 were considered statistically significant.

## 3. Results

### 3.1. Demographic and Clinical Characteristics of the Study Sample

The demographic data of the subjects included in the study are shown in Table 1. The median age was 9 years for each group (the youngest being 5 years and the oldest 16 years), showing no significant differences between the group of children with disabilities and the group of children without disabilities (*p* = 0.6986).

In terms of gender, the group of children without disabilities and special health needs enlisted a higher number of girls than boys, although the difference was statistically insignificant (*p* = 0.9302). All the children who took part in the study were of white European ancestry.

Regarding residence, the majority of children in each group lived in urban areas (*p* = 0.6834). The average age of the mother at childbirth was higher for the group with special health needs, but this difference was statistically insignificant (*p* = 0.1010).

### 3.2. Visual Impairment and Passive Exposure to Smoking during Childhood

The percentage of children passively exposed to present and past smoking around their mothers was marginally higher in the group of children with special health needs than in the control group but did not show any statistical significance (*p* = 0.3453). Another variable taken into consideration was the percentage of children whose mothers never smoked. This situation was less common in the group of children with disabilities than in the control group but was still statistically insignificant (Table 1).

The percentages of children whose fathers never smoked and of children exposed to former passive smoking from their fathers were also marginally higher in the group of children with special health needs (*p* = 0.2426). The percentage of children whose fathers currently smoke was insignificantly higher in the control group (*p* = 0.2426) (Table 1).

Another demographic factor taken into account was the wearing of glasses by the parents. The cases in which the mother wore glasses were similar between both groups of children (*p* = 0.5190). The groups were similar in terms of demographic factors, except for the number of cases where the father wore glasses, which was significantly more common among the pediatric patients with special health needs than among the children without special health needs (*p* = 0.0047) (Table 1).

### 3.3. Clinical Parameters

Table 2 provides certain clinical parameters regarding the visual acuity, refractive status, and plasmatic vitamin D levels of the children. The initial median visual acuity (IQR) was statistically significantly lower in the group of children with disabilities than in the control group (*p* < 0.0001). The children with special health needs presented an initial average visual acuity (IQR) of 0.525 (with values between 0.15 and 1), while those without disabilities had an IQR of around 1 (with values between 0.87 and 1).

The groups of children were also comparable in terms of their emmetropic status of the right eye (RE) and left eye (LE). This aspect was statistically significantly less common in the group of children with special health needs (*p* < 0.0120). The percentage of ametropia was statistically significantly higher in the group of children with special health needs (*p* < 0.0120).

The final average visual acuity (IQR) was statistically significantly lower for those with disabilities than for those in the control group (*p* < 0.0001). The table of clinical features indicates more severe impairment of visual acuity among patients with special health needs in the group we ophthalmologically examined. The initial average visual acuity of both eyes was, moreover, significantly lower for the children with special health needs. The improvement of final visual acuity was more efficient for children without special health needs (in terms of the average score, *p* < 0.0001).

Plasma vitamin D levels showed significant differences between groups, with lower levels observed in children with disabilities (*p* < 0.0001).

### 3.4. The Relative Risk of Visual Impairment in Children with Special Health Needs

Visual impairment is defined in our study as an average initial visual acuity below 0.8: (VA RE + VA LE)/2 < 0.8. The incidence of visual impairment in the two study groups showed a relative risk that was 2.8636 times higher (95% CI: 1.44–5.67; *p* < 0.0026) for children with disabilities to develop visual impairments compared to the control group (Table 3). An initial average visual acuity of >0.8 for both eyes was observed in 44 children in the group with disabilities and in 63 children in the control group. Therefore, visual impairment (an average initial visual acuity <0.8) was found to be more frequent in the group of children with special health needs (statistically significant, *p* < 0.0001), as shown in Table 3.

### 3.5. The Correlation between Plasma Vitamin D Levels and Initial Average Visual Acuity

The correlation between plasma vitamin D levels and initial average visual acuity for both eyes was calculated together for all the children in this study. The results confirmed a close correlation: Spearman’s coefficient (rho) = 0.612; 95% CI: 0.505–0.700; *p* < 0.0001. The close correlation between plasma vitamin D levels and low average initial visual acuity demonstrates that the lower the plasma vitamin D level, the lower the visual acuity. The graphical representation in Figure 1 includes all the patients in this study as correlation points between the two variables.

### 3.6. Independent Risk Factors for the Development of Visual Impairments in the Studied Group of Patients

The independent risk factors for the development of visual impairments in our study group were the following: gender, area of residence, mother’s age at birth, parental smoking, parental wearing of glasses, presence of a disability, and plasma vitamin D levels. Among these features, only the following proved to be independent risk factors (Table 4): the mother’s wearing of glasses, the mother being a past smoker, the mother being a present smoker, and plasma vitamin D levels.

All the children were pooled into one group, allowing us to use a logistic regression test with gradual entry to analyze all the risk factors for the development of visual impairment. The logistic regression test found that the presence of a disability was not an independent risk factor for the development of visual impairment, while both a low plasma vitamin D level and the mother’s history of tobacco use and wearing of glasses significantly influenced the appearance of visual impairment. The greatest risk was observed for children who were exposed in the past to their mothers’ smoking habits.

## 4. Discussion

Early identification of visual impairment and the correction of refractive errors might help overcome barriers in the general and academic development of children, as well as improve the overall quality of life for both children who are clinically healthy and those who have special health needs. Ophthalmological evaluations of children with special health needs must be done as they would be for clinically healthy children, even though such evaluations can be challenging due to the adaptations needed for certain children. Children no younger than 5 years old were chosen to participate in this study because younger children can be easily distracted by the environment.

The study showed impairment of initial visual acuity among the patients with special health needs in the group we ophthalmologically examined, and the initial average visual acuity (without correction) of both eyes was lower for the children with disabilities. The improvement of the final visual acuity was more efficient for the children without special health needs (*p* < 0.0001). These results highlight the need to engage in ophthalmological screening of children with disabilities, as such children present a higher prevalence of visual impairment for the initial and final measurements (*p* < 0.0001) and a relative risk of 2.8636 times higher (*p* < 0.0026) of developing visual impairments compared to the control group of clinically healthy children. The mean final visual acuity in our group was higher than the mean final visual acuity of 0.349 (range 0.06 to 0.63) determined by Mocanu for 35 children aged 5–16 years in the south-west of Romania, Timisoara. There were no data provided on any disabilities among those children [6]. A study conducted by Woodhouse on 166 pupils with special needs in Wales, UK, with a mean age of 12 years, revealed 29 of them to have visual impairment [13].

Visual impairment was detected in 36 children with special health needs in our study, which is fewer than the 52 cases identified by Bezabih in a study conducted in Ethiopia on 718 disabled students aged 6–18 years old [15].

Children with special health needs can be genetically prone to refractive errors, with refractive errors often being the most prevalent ocular morbidity among those children [14]. The risk of refractive errors was reported by Vora to be significantly higher in children with special needs (relative risk, 48.1 (95% CI: 17.54–131.8)) [3]. When we compared the prevalence of refractive errors between groups, we observed a significant prevalence of 55% refractive errors for the RE and LE in the group of children with disabilities, which is higher than the 29% prevalence of refractive errors in the RE and LE identified by Fletcher and Thompson among 102 children with disabilities aged <18 years old from the USA [16]. A study conducted by Nielsen on 4–15-year-old children with special health needs from Copenhagen revealed a 39% prevalence of refractive errors [17]. A similar result to ours was observed by Kumar in India; the authors found a 54.5% prevalence of refractive errors in a study group of 116 children with different disabilities [18]. This result suggests that refractive errors are relatively common among children with special health needs, so this pathology must be identified and treated accordingly. Our data correlate with other data found in the literature, namely the 58.5% prevalence of refractive errors in children with special health needs reported by Vora in Oman and the 49% prevalence of refractive errors found among 137 children with disabilities identified by Bankes in the UK [3,19].

We analyzed the association between bilateral decreased visual acuity and different factors in a base sample population of children aged 5–16 years old. The logistic regression provided important insights regarding the major risk factors associated with decreased bilateral VA. The risk factors were the mother being a former smoker, the mother wearing glasses, the mother being a present smoker, and the child’s serum level of vitamin D. This study did not include the ages of the children as a possible risk factor for developing visual impairment or the residences of the children. Similarly, Gogate did not identify a link between the age of the child and the appearance of a form of visual impairment [20]. We found a correlation between a child’s visual impairment and the mother’s smoking but not the father’s smoking. In the literature, Lin L.L. reported similar findings that do not show a correlation between the presence of visual impairment in children and their fathers’ smoking habits. The study was conducted between 1999 and 2001 on children of Chinese origin [21].

We identified one Romanian study by Mocanu that showed a mother who previously smoked to be a risk factor for the visual impairment of her children [6]. The association between maternal smoking and the visual impairment of children was also demonstrated by the Avon Longitudinal Study of Parents and Children (ALSPAC) of 2008. This Australian study raised awareness of the association between maternal smoking and visual impairment among children (OR: 1.4; 95% CI: 1.0–1.9) [22]. Li revealed that children who were passively exposed in the past to cigarette smoke from their mothers had a 1.47-fold higher risk of developing visual impairment than those who were not exposed to smoke [23].

Our results show that children who are passively exposed to maternal smoking are 10.4079 times more prone to develop some form of visual impairment (95% CI: 2.1923 to 49.4115). A child who is still being exposed to passive smoking is 4.5238 times more likely to develop a visual deficiency (95% CI: 1.1371 to 18.0695).

However, as this is a retrospective study, we cannot provide firm conclusions on the relationship between cigarette smoke and the emergence of visual impairment. We can only state that the link between visual impairment and passive exposure to smoking is dependent upon the period of exposure, the amount of exposure, and the sensitivity of the brain’s nicotine receptors. Thus, the higher risk rate observed for children whose mothers smoked in the past may be due to the longer periods of time those children were exposed to passive smoking. Furthermore, this study has underscored the need for future, more detailed research on the nicotine receptors in the brain to reveal possible linkages with the development of visual impairments. Data from the literature attesting the possible correlation between the mother’s smoking habits and the visual impairment of the child indicated different results. The Twins Eye Study in Tasmania reported a significant correlation between a mother’s smoking habits and the appearance of visual impairment in her child. The study was conducted on a group of 346 children (167 pairs of twins and 4 sets of triplets) with an average age of 9.25 ± 2.4 years [24]. Exposure to smoking has many adverse effects, so this study demonstrates that children’s passive exposure to cigarette smoke should at least be limited, if not avoided altogether.

The children in this study were pooled together as a whole group for the multivariate analysis of the relative risk of visual impairment based on the mother wearing glasses. We observed a relative risk of 8.8340 (95% CI: 2.4066 to 3.4275) to develop visual deficiency among the children whose mothers wear spectacles. Similar results were found by Ezhilvathani N in a study conducted on 200 children. The results positively associated a family history of wearing glasses to the inheritance of refractive errors and visual impairment in children [25].

When we further evaluated the relationship between plasma vitamin D levels and visual acuity, we found a significant correlation (*p* < 0.0001), indicating the lower the plasma vitamin D level, the lower the visual acuity. Previous studies conducted by Dayna S on 2392 people aged 21–84 concluded that there is no correlation between visual acuity and low levels of plasma vitamin D [26]. However, vitamin D has a major role in the overall state of one’s health, and vitamin D deficiency was given epidemic status in the USA, as the impact of such a deficiency at the retinal level is rapid [27]. Vivian Lee’s study observed that the administration of vitamin D for six weeks on mice had an impact on the cones, improving retinal function and visual function overall [28]. McMillan J stated that the eye is fully dependent on vitamin D, whose receptors are found in the cornea and lens and thus mediate the refractive status of the eye, as well as visual acuity as a secondary function [11]. McMillan also stated that some degree of refractive status and visual acuity can be reversed under the adequate replacement of vitamin D [11]. These observations should be explored further and confirmed by controlled longitudinal studies.

The strengths of our study include its retrospective nature and its focus on children with disabilities, compared to focusing on a control group consisting of clinically healthy children without any chronic conditions. However, our study has some limitations. The first is that the sample comprised only small groups of children with and without disabilities; a larger sample size for each group would have increased the strengths of the study. Second, the correlation of visual impairment with vitamin D level was observational and could not indicate a causality. Our investigation did not allow us to draw firm longitudinal conclusions or to generalize the results to the entire child population in Romania. Randomized trials are needed to find a causative factor. This study will help raise awareness about the status of children’s eyes—for both clinically healthy children and children with special health needs—and to plan national clinical services for detecting visual impairment in children.

## 5. Conclusions

To the best of our knowledge, this is the first study investigating the refractive status and visual acuity of children with disabilities in the north-western part of Romania. Children with special health needs are much more prone to refractive errors and visual impairment than their clinically healthy peers. The major risk factors for a child’s visual impairment are exposure to maternal smoking, his or her mother wearing glasses, and the plasmatic level of vitamin D.

Knowledge regarding the prevalence of refractive errors and the modifiable risk factors for visual impairment in children with special health needs must be determined and understood for these children to receive optimal support.

## Figures and Tables

**Figure 1 medicina-57-00403-f001:**
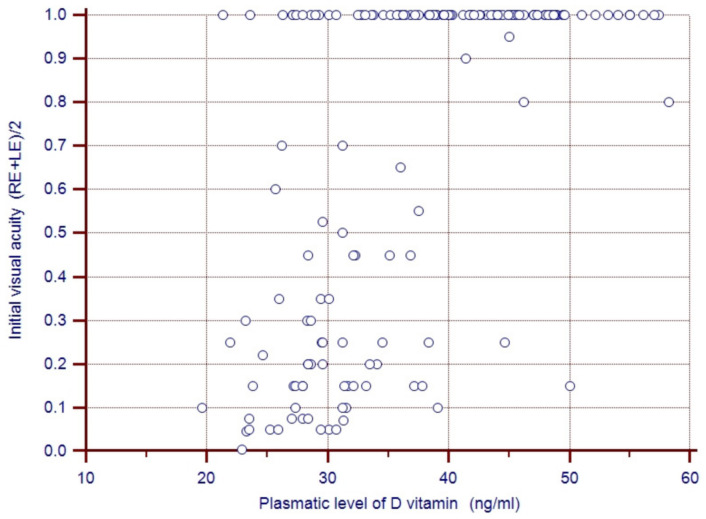
Correlation between plasmatic vitamin D level and mean initial visual acuity for both eyes in the whole group.

**Table 1 medicina-57-00403-t001:** Comparative table of the demographic characteristics of the two study groups.

Demographic Characteristics	Group of Children without Disabilities *n* = 81	Group of Children with Disabilities *n* = 80	Statistical Significance (*p*)
Age (years)—median (IQR)	9 (7–12)	9 (6–12)	0.6986 *
Gender M/F	36/45	36/44	0.9302 **
Residence U/R	44/37	47/33	0.6834 **
Mother’s age at birth of child (years)—average (±SD)	27.7 (5.7)	29.4 (6.8)	0.1010 ***
Smoking (mother)			
- former (%)	9 (11.1)	12 (15)	0.3453 ****
- present (%)	10 (12.3)	15 (18.8)	
- never (%)	62 (76.6)	53 (66.2)	
Smoking (father)			
- former (%)	7 (8.6)	9 (11.3)	0.2426 ****
- present (%)	29 (35.8)	19 (23.7)	
- never (%)	45 (55.6)	52 (65)	
Mother wears glasses Yes/No	13/68	17/63	0.5190 **
Father wears glasses Yes/No	11/70	27/53	<0.0047 **

*n* = number of patients; IQR = interquartile range; M = male; F = female; U = urban; R = Rural; SD = standard deviation. * Mann–Whitney test; ** chi-square test with Yates’ correction; *** Student’s *t*-test for independent groups; **** chi-square test. Statistically significant *p* values are highlighted in bold.

**Table 2 medicina-57-00403-t002:** Comparative table of clinical parameters between the two study groups.

Clinical Features	Group of Children without Disabilities *n* = 81	Group of Children with Disabilities *n* = 80	Statistical Significance (*p*)
Initial VA—average (IQR) uncorrected	1 (0.87–1)	0.525 (0.15–1)	<0.0001 *
RE modification			
- Emmetropia	61 (75.3)	36 (45)	0.0120 **
- Ametropia (%)	20 (24.7)	44 (55)	
LE modification			
- Emmetropia	62 (76.5)	36 (45)	0.0035 **
- Ametropia (%)	19 (23.5)	44 (55)	
Final VA—average (IQR)	1 (1–1)	1 (0.675–1)	0.0001 *
Plasma vitamin D level (ng/mL)—average (±SD)	42.25 (8.1)	32.52 (7.3)	<0.0001 ****

*n* = number of patients; IQR = interquartile range; VA = visual acuity; RE = right eye; LE = left eye. Data are expressed as the mean ± standard deviation, unless otherwise indicated. * Mann–Whitney test; ** chi-square test; **** Student’s *t*-test for independent groups. Statistically significant *p* values are highlighted in bold.

**Table 3 medicina-57-00403-t003:** Incidence of visual impairment (average initial VA < 0.8) in the two study groups.

Average Initial VA for Both Eyes	Group of Children without Disabilities *n* = 81	Group of Children with Disabilities *n* = 80	Statistical Significance (*p*)
Without visual impairment (average VA ≥ 0.8)	63	44	<0.0001 *
With visual impairment (average VA < 0.8)	18	36	<0.0001 *

*n* = number of patients; VA = visual acuity tested with LogMar; * chi-square test with Yates’ correction. Statistically significant *p* values are highlighted in bold.

**Table 4 medicina-57-00403-t004:** Independent risk factors for developing visual deficiency in the studied group of patients.

Risk Factors	Relative Risk	CI 95%
Mother wears glasses	8.8340	2.4066 to 3.4275
Mother being a former smoker	10.4079	2.1923 to 49.4115
Mother being a present smoker	4.5328	1.1371 to 18.0695
Plasma vitamin D level	0.7603	0.6935 to 0.8336

## Data Availability

The data presented in this study are available on request from the correspondence authors.

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
