# Peer review of "Observations on Refractive Status and Risk Factors for Visual Impairment in Children with Disabilities"

_medicina, 2021, doi:10.3390/medicina57050403_

Round 1

Reviewer 1 Report

This manuscript has an important clinical message, and should be of great interest to the readers. The literature review is thorough, with important points and well analyzed.

I have only two considerations in order to improve the manuscript:

  1. Lines 210-211. Why do you use this definition of Visual impairment? What is it based on?
  2. Table 1. In this table there are some data messy in “smocking father” point.

Author Response

Hello!

  1. Lines 210-211. Why do you use this definition of Visual impairment? What is it based on?

We chose this limit of 0.8 visual acuity to be the limit in our study where visual impairment starts because the International Council of Ophthalmology states that vision loss begins at this limit of 0.8 and that it is suitable for clinical studies. We have more children under this limit in each group, so we chose to compare the groups under this value. 

2. Table 1. In this table there are some data messy in “smocking father” point.

We reorganized the table. Thank you for your observations!

We hope that in this form, the quality of the article is improved.

Thank you for agreeing with the rest!

Reviewer 2 Report

This article dealt with the prevalence of refractive errors and visual impairment in children with disabilities, compared to healthy children, also exploring the risk factors for visual impairment in this population.

There are some points of this article that need to be improved and clarified.

1) In the Abstract, the authors should indicate the number of children included in each study group, thus providing an important and immediate information for the readers.

2) In the Introduction section, after the background, the authors should explain the aim of their study, also citing the main investigations performed. Moreover, the authors should better explain the rationale of the study, discussing and referencing why, in their study, they looked for parents’ smoking and wearing glasses as risk factors for visual impairment in children, as done for plasma vitamin D level.

3) In the Materials and Methods section, the authors stated that “parents of the children were asked for dosing vitamin D for their children at any local laboratory desired”. May this create a bias in the study, since each laboratory has its own reference range values? Authors should clarify this and possibly cite it as a study limitation.

4) Which scale (logMAR, decimal) was used to test visual acuity? The authors should specify this information. 

5) Although a substantial number of children were enrolled in the study, a statistical power assessment would be appropriate and of interest.

6) In the Statistical analysis, the authors stated that “correlations were evaluated using Spearman coefficients, as appropriate”. This means that all the analyzed data for correlations have no normal distribution? The authors should clarify this information.

7) In the Results section, the authors should avoid to repeat in the main text what has already been explained in the Tables.

8) For Spearman’s coefficient, in addition to p-value, the authors should also provide 95% confidence interval to strengthen the significance of the data.

9) In the Figure 1, the authors should add the regression line for a more complete explanation of the data.

10) The authors, in the Discussion section, stated that “the strengths of our study consist in the fact that it is a prospective one, conducted on children with disabilities, compared to a control group consisted of clinically healthy children, without any chronic condition”.

Considering this study, it seems to be a case-control study, which is usually retrospective, not prospective. Furthermore, the retrospective nature of this study is also suggested by the fact that the authors performed only one ophthalmological evaluation, without any other follow up examinations. In addition, any potential damage or effect of disabilities on visual acuity and/or refractive errors has already occurred at the time of the ophthalmological evaluation, also suggesting a retrospective nature of the study.

The authors should think about this aspect, clarifying the retrospective or prospective nature of the study.

11) The English should be improved throughout the manuscript.

Author Response

Our responses are highlighted in red, as in the text of the manuscript the changes/corrections requested were highlighted in red.

1) In the Abstract, the authors should indicate the number of children included in each study group, thus providing an important and immediate information for the readers.

Thank you very much for your suggestion! We completed this section by adding data regarding the number of children included in each study group. 

2) In the Introduction section, after the background, the authors should explain the aim of their study, also citing the main investigations performed. Moreover, the authors should better explain the rationale of the study, discussing and referencing why, in their study, they looked for parents’ smoking and wearing glasses as risk factors for visual impairment in children, as done for plasma vitamin D level.

Thank you very much for your observations! We explained the aim of our study in the Introduction section. We hope that in this form, the quality of the article is improved.

3) In the Materials and Methods section, the authors stated that “parents of the children were asked for dosing vitamin D for their children at any local laboratory desired”. May this create a bias in the study, since each laboratory has its own reference range values? Authors should clarify this and possibly cite it as a study limitation. Thank you very much for the observation! All the local laboratories had the same reference values. 

4) Which scale (logMAR, decimal) was used to test visual acuity? The authors should specify this information. 

Thank you again for the observation! We used decimal scale to test visual acuity.

5) Although a substantial number of children were enrolled in the study, a statistical power assessment would be appropriate and of interest.

Thank you very much! For a type 2 error (β) of 0,20 the minimum number of patients should be 20. For an error of type2 0,05 the minimum number pf patients should be 30. We had a bigger number of children included in the study, so the statistical power of the study is bigger.

6) In the Statistical analysis, the authors stated that “correlations were evaluated using Spearman coefficients, as appropriate”. This means that all the analyzed data for correlations have no normal distribution? The authors should clarify this information.

Vitamin D level is with normal distribution and the initial visual acuity is with asymmetric distribution (skewed), so that the Spearman test was used for the correlation of these two variables.

7) In the Results section, the authors should avoid to repeat in the main text what has already been explained in the Tables.

Thank you very much for the pertinent observation! We reformulated the main text.

8) For Spearman’s coefficient, in addition to p-value, the authors should also provide 95% confidence interval to strengthen the significance of the data.

We added the 95% CI in the main text. It is 95% CI: 0.505-0.700

 9) In the Figure 1, the authors should add the regression line for a more complete explanation of the data.

We wanted a stronger correlation between plasmatic vitamin D level and the initial visual acuity of all 161 children included in this study. The regression line can be done only if we eliminate the children with initial visual acuity of 1.

10) The authors, in the Discussion section, stated that “the strengths of our study consist in the fact that it is a prospective one, conducted on children with disabilities, compared to a control group consisted of clinically healthy children, without any chronic condition”.

Thank you for the observation! The study conducted was a retrospective one. Children were evaluated only once.

11) We recalculated the relative risk of visual impairment and it is 2.8636 (95% CI: 1.44-5.76, P<0.0026).

All the data were introduced in the manuscript, highlighted in red.

Thank you very much for all your observations made for the study and article! They are pertinent for us and made us improve our style of thinking about the study and article and will also be very important for all of our future projects. Thank you very much!

Round 2

Reviewer 2 Report

Although most of the criticisms have been addressed, the authors should not consider the retrospective nature of their study as a strength, but as a limitation.

Furthermore, English needs to be improved.

Author Response

Comments and Suggestions for Authors

Although most of the criticisms have been addressed, the authors should not consider the retrospective nature of their study as a strength, but as a limitation.

Thank you very much for your observation! We adressed this situation about the nature of the study and we made the required changes, highlighted in red, in the article.

One strength of this study is its focus on children with disabilities, compared to focusing on a control group consisting of clinically healthy children without any chronic conditions. The current study is subject to some limitations. First of all, it is a retrospective case-control study.  

Round 3

Reviewer 2 Report

The authors addressed all the criticisms of the study.